# The Uncertain Benefit of Adjuvant Chemotherapy in Advanced Low-Grade Serous Ovarian Cancer and the Pivotal Role of Surgical Cytoreduction

**DOI:** 10.3390/jcm10245927

**Published:** 2021-12-17

**Authors:** Racheal Louise Johnson, Alexandros Laios, David Jackson, David Nugent, Nicolas Michel Orsi, Georgios Theophilou, Amudha Thangavelu, Diederick de Jong

**Affiliations:** 1ESGO Center of Excellence in Advanced Ovarian Cancer Surgery, Department of Gynaecological Oncology, St. James’s University Hospital, Leeds LS9 7TF, UK; a.laios@nhs.net (A.L.); david.nugent@nhs.net (D.N.); georgios.theophilou@nhs.net (G.T.); amudha.thangavelu@nhs.net (A.T.); diederick.dejong@nhs.net (D.d.J.); 2Department of Medical Oncology, St. James’s University Hospital, Leeds LS9 7TF, UK; davidp.jackson@nhs.net; 3Leeds Institute of Medical Research, St. James’s University Hospital, Leeds LS9 7TF, UK; N.M.Orsi@leeds.ac.uk

**Keywords:** low-grade serous ovarian cancer (LGSOC), high-grade serous ovarian cancer (HGSOC), cytoreduction, surgical complexity, survival

## Abstract

In our center, adjuvant chemotherapy is routinely offered in high-grade serous ovarian cancer (HGSOC) patients but less commonly as a standard treatment in low-grade serous ovarian cancer (LGSOC) patients. This study evaluates the efficacy of this paradigm by analysing survival outcomes and by comparing the influence of different clinical and surgical characteristics between women with advanced LGSOC (*n* = 37) and advanced HGSOC (*n* = 300). Multivariate analysis was used to identify independent prognostic features for survival in LGSOC and HGSOC. Adjuvant chemotherapy was given in 99.7% of HGSOC patients versus in 27% of LGSOC (*p* < 0.0001). The LGSOC patients had greater surgical complexity scores (*p* < 0.0001), more frequent postoperative ICU/HDU admissions (*p* = 0.0002), and higher peri-/post-operative morbidity (*p* < 0.0001) compared to the HGSOC patients. The 5-year OS and progression-free survival (PFS) was 30% and 13% for HGSOC versus 57% and 21.6% for LGSOC, *p* = 0.016 and *p* = 0.044, respectively. Surgical complexity (HR 5.3, 95%CI 1.2–22.8, *p* = 0.024) and complete cytoreduction (HR 62.4, 95% CI 6.8–567.9, *p* < 0.001) were independent prognostic features for OS in LGSOC. This study demonstrates no clear significant survival advantage of chemotherapy in LGSOC. It highlights the substantial survival benefit of dynamic multi-visceral surgery to achieve complete cytoreduction as the primary treatment for LGSOC patients.

## 1. Introduction

Ovarian cancer is the fourth most common cause of cancer-related death in women worldwide and is the most lethal gynecological malignancy [1]. Since efficient and cost-effective screening tools for ovarian cancer are unavailable [2,3], efforts have been focused on optimizing treatment strategies.

Epithelial ovarian cancer (EOC) is categorized by diverse histological subtypes and differentiations [4]. Serous epithelial ovarian cancer is the most common subtype and is no longer considered as a homogenous disease but has instead been divided into two distinct tiers: low-grade serous ovarian cancer (LGSOC) and high-grade serous ovarian (HGSOC) [5]. Differences between these two groups extend beyond histological grading and involve contrasting genetic profiles, aetiology, and clinical response to treatment. HGSOC represents over 90% of all serous ovarian carcinomas and is characterized by rapid progression. LGSOC is a rarer sub-type, accounting for 5–10% of the serous epithelial tumour population and possesses an indolent nature, which is associated with improved survival outcomes [6,7]. Responses to chemotherapy regimens are poor in the neo-adjuvant [8] as well as in the recurrent setting [9] for LGSOC. Standard management practices for both of these groups have historically been identical and consist of cytoreductive surgery with systemic platinum-based chemotherapy [10,11]. Data regarding the omission of systemic adjuvant treatment following cytoreductive surgery in advanced-stage LGSOC is not widely available. Two randomized control, phase III trials are currently under way to examine the role of systemic treatment in LGSOC [12,13].

Cytoreductive surgery, which aims at nil macroscopic residual disease (RD), is a pivotal part of treatment for ovarian cancer [14,15]. The efficacy and optimal timing of surgery across the various subtypes of advanced EOC are less clear [16]. Different treatment regimens have been developed using this approach. Patients are treated with neoadjuvant chemotherapy (NACT) followed by interval debulking surgery (IDS) when unacceptable surgical morbidity is expected, or complete cytoreduction is unlikely to be achieved [17]. Treatment with this regimen has been associated with similar outcomes compared to primary debulking surgery (PDS) followed by adjuvant chemotherapy [18,19]. However, such studies generally combine various tumour subtypes in their analyses, with low-grade tumours representing 3.3–7.8% of the studied population [20,21,22]. Low-grade and high-grade tumours have different characteristics, and therefore, the efficacy of surgical management in these tumours may not be comparable [11]. Consequently, possible differences in outcomes (if any) may have been skewed due to the representation of these subgroups in the studied population.

This study was primarily designed to assess the differences in the clinical management strategies between patients with advanced LGSOC and those with HGSOC. The second aim of this study was to evaluate the justification of omitting standard adjuvant chemotherapy in LGSOC. By comparing the surgical characteristics and survival outcomes between low- and high-grade tumours, we hypothesised that there would be a disparity in the treatment characteristics and survival outcomes.

## 2. Materials and Methods

### 2.1. Selection of Patients and Study Design

Prospectively collected data from a cohort of consecutive patients who had been treated for advanced-stage serous EOC between January 2014 and December 2017 in St. James’s University Hospital, Leeds, United Kingdom, were retrieved from the hospital wide electronic patient database PPM [21]. Treatment and follow-up data were collected until July 13th, 2021. The eligibility criteria were patients with International Federation of Gynecology and Obstetrics (FIGO) stage III-IV EOC who had cytoreductive surgery, either in the upfront or neo-adjuvant setting [22]. All patients underwent surgical cytoreduction by a certified consultant in Gynecological Oncology. Patients with grade two EOC, those patients with a synchronous primary malignancy (*n* = 1) or recurrent ovarian malignancy (*n* = 5), and those who had surgery in an emergency setting (*n* = 3) were excluded, see Figure 1. This study was approved by the institutional review board (ID 282396); informed consent was waived, and the study was performed according to the standards outlined in the Declaration of Helsinki.

### 2.2. Workup, Chemotherapy, and Surgical Procedure

For this analysis, age was defined as age at the time of diagnosis. The clinical condition of the patients was scored by means of the performance status (PS) [23]. All of the patients underwent a pre-treatment physical examination, serum CA 125 measurement, chest, abdomen, and pelvis imaging by CT-scanning, and histological diagnosis by either image-guided or surgical biopsy. The results of the pre-treatment workup were discussed in our multi-disciplinary team (MDT), which recommended upfront surgical cytoreduction with or without subsequent adjuvant chemotherapy or NACT followed by surgical cytoreduction with or without adjuvant chemotherapy.

Patients receiving NACT followed by surgical cytoreduction were those patients in which unacceptable surgical morbidity was expected, such as patients with PS > 2 or those requiring extensive preoperative optimization or those in who complete cytoreduction was unlikely to be achieved, as indicated by imaging. NACT consisted of three courses of combined carboplatin and paclitaxel chemotherapy prior to surgery with or without a subsequent three courses of adjuvant chemotherapy. A total of 13 LGSOC patients received NACT (35.1%) compared to 240 HGSOC patients (80%). Adjuvant chemotherapy in the upfront surgical cytoreduction setting usually consists of six courses of combined carboplatin and paclitaxel chemotherapy. Patients with LGSOC were offered adjuvant combined chemotherapy but were counselled regarding the chemo-resistant nature of the disease and therefore the limited lack of efficacy. LGSOC patients who had no radiological response from NACT (*n* = 6) or who had radiologically progressed on NACT (*n* = 2) were counselled against adjuvant chemotherapy due to lack of benefit. A total of seven LGSOC patients received post-operative adjuvant chemotherapy only, four patients received six cycles of carboplatin and paclitaxel, and three patients received single agent carboplatin. A further three LGSOC patients received post-operative chemotherapy as part of their interval debulking regime. Other adjuvant management strategies for this LGSOC cohort included bevacizumab in combination with chemotherapy (*n* = 1), maintenance letrozole (*n* = 2), and maintenance tamoxifen (*n* = 2).

Surgical cytoreduction was performed by an abdominal midline incision followed by an assessment of disease dissemination in the abdomen according to the peritoneal cancer index (PCI) [24]. The procedure was characterized by sampling any ascitic fluid, total hysterectomy, bilateral salpingo-oophorectomy, and omentectomy as the bare minimum. To achieve a complete surgical cytoreduction, the procedure could be extended by stripping the diaphragm and peritoneum, stripping the mesentery, wedge resection of the liver, (partial) gastrectomy, cholecystectomy, splenectomy, pancreas tail resection, adrenalectomy, small and/or large bowel resection with or without stoma formation, appendicectomy, and lymph node dissection. On rare occasions, lateral extended endopelvic resection (LEER) or composite exenteration were required to achieve the desired surgical result [25,26].

### 2.3. Primary and Secondary Outcome Parameters

The primary outcome parameter of overall survival (OS) was calculated from the date of diagnosis to the date of death from the disease. The secondary parameter, progression-free survival (PFS) was calculated from the date of diagnosis to the date of confirmed recurrence. Other secondary parameters included the timing of surgical cytoreduction (PDS or IDS), PS, complexity of the surgical procedure, residual disease (RD), duration of the surgical procedure, intra-operative blood loss, utilization of the high-dependency (HDU) or intensive care unit (ICU), peri- and post-operative morbidity, and length of hospital stay. Residual disease was categorized according to the size of the remaining tumour nodules at the end of the surgical procedure. Complete cytoreduction was defined as a no measurable macroscopic RD (CC 0) or RD < 2.5 mm (CC 1). Incomplete cytoreduction was defined as CC 2 (2.5 mm ≥ RD < 2.5 cm) or CC 3 (RD ≥ 2.5 cm) [24]. The complexity of the procedure was scored according to the surgical complexity score (SCS) [27]. Peri- and postoperative morbidity and mortality were classified according to the Clavien–Dindo (CD) scoring system [28]. Follow-up was at regular intervals over five years.

### 2.4. Staging and Tumour Assessments

The disease was staged as defined by the 2014 FIGO criteria [20]. Differentiation was classified as being low grade and high grade according to the two-tier classification system [5]. Data regarding the size and the location of the postoperative residual tumours and the histopathological features were collected.

### 2.5. Statistical Analysis

The characteristics of the patients according to group were presented as means +/− SD. Differences between the groups were analyzed by the Chi-squared, Student T-test, or Mann–Whitney tests, depending on the data distribution. Standard univariate analysis compared survival curves using the Kaplan–Meier method, and the statistical significance of these parameters were calculated using the log-rank test. Multivariate analyses were performed using Cox’ proportional hazard method. Independent variables that were found to have a *p* value < 0.1 in the univariate analysis were then combined in a multivariate analysis. All of the tests were two-sided, and *p* < 0.05 was considered significant. The software packages GraphPad Prism (GraphPad Software, San Diego, CA, USA) and SPSS 27 (SPSS, Chicago, IL, USA) were employed for data analysis.

## 3. Results

### 3.1. Patients

A total of 337 patients were enrolled in the study: 37 confirmed LGSOC and 300 patients with HGSOC. There were no statistical differences in age, FIGO stage, or performance status between the two groups. The median pre-treatment CA125 measurements were 122 (range 25–9657) U/mL for the LGSOC patients versus 875 (range 13–28,600) U/mL for the HGSOC patients, *p* < 0.0001. The pre-treatment CT scan showed calcified deposits in 78.4% of the LGSOC patients compared to in 5.7% of the HGSOC patients, *p* < 0.0001. Details are displayed in Table 1.

### 3.2. Treatment Parameters

Primary cytoreduction was performed in 64.9% of the patients with LGSOC versus in 20.0% of the patients with HGSOC, *p* < 0.0001. The median PCI in the patients with LGSOC was eight (range 2–21) versus five (range 1–24) for those with HGSOC, *p* = 0.0216. A complete surgical cytoreduction (CC 0–1) was achieved in 89.2% of the LGSOC patients and in 88.0% of the HGSOC patients, *p* = 0.833. Moderate and high SCS were found in 59.5% and 8% for the LGSOC group versus in 24.6% and 4.4% for the HGSOC group, *p* < 0.0001. On average, cytoreductive surgery lasted 207 ± 93 min in the LGSOC patients and 150 ± 63 min in the HGSOC patients, *p* = 0.0019. The average intraoperative blood loss for patients with LGSOC was 632 ± 329 cc compared to 478 ± 323 cc for those with HGSOC, *p* = 0.0002. An elective postoperative ICU/HDU admission was required in 37.8% of the patients with LGSOC versus in 16.3% of the patients with HGSOC, *p* = 0.0015. The median hospital stay length for the LGSOC patients was 9 (range 4–30) days, and it was 7 (range 7–68) days for the HGSOC patients, *p* = 0.0002. Median end of treatment CA125 in the patients with LGSOC was 17 (range 5–139) U/mL versus 13 (range 3–4019) U/mL in the HGSOC, *p* = 0.215. Clinically relevant perioperative morbidity, CD 3A or higher, was observed in 27% of the patients with LGSOC compared to in 6.6% of those with HGSOC, *p* < 0.0001. One patient with HGSOC died within 30 days following cytoreductive surgery due to bowel related complications. Two further patients died 60 days postoperatively, one in each group. Neither were directly related to their surgical procedure; one death was related to chemotherapy complications at 48 days, and the other died from an undisclosed accident at 45 days, respectively. The majority of patients with LGSOC (73%) did not receive adjuvant chemotherapy treatment, in contrast to only one HGSOC patient (0.3%) *p* < 0.0001. Details are displayed in Table 2.

### 3.3. Survival Outcomes

The median follow-up was 70 months. The 5-year OS for the whole cohort was 33% (median OS 41 months 95% CI 35.3–46.7). The 5-year OS for patients with LGSOC was 57% (median not reached mean OS 59.7 months, 95% CI 50.5–69) compared to 30% (median 40 months 95% CI 34.6–45.4) for those with HGSOC, *p* = 0.016 (Figure 2). The five-year PFS for patients with LGSOC was 21.6% (median 22 months 95% CI 9.2–34.8) versus 13% (median 17 months 95% CI 15.3–18.7) for the group with HGSOC, *p* = 0.044.

### 3.4. Impact of Surgical Approach on Survival Outcomes

A superior OS was established in the entire cohort with PDS when compared to those with IDS (Figure 3). The 5-year OS and PFS for the PDS patients was 51% and 16% (*p* = 0.001) versus 24% and 11% for the IDS patients (*p* = 0.001). The median OS for the LGSOC patients who underwent PDS was not reached, mean 68.6 months (95% CI 58.8–73.4) and median PFS 29 months (95% CI 6.2–51.8) versus 66 months (95% CI 48.6–83.5) and 24 months (95% CI 18.9–29) for those HGSOC patients who underwent PDS, *p* = 0.097 and 0.552, respectively. The median OS and PFS for the LGSOC patients who underwent IDS was 31 months (95% CI 18–44) and 16 months (95% CI 11.2–20.7) compared to 36 months (95% CI 31.6–40.4) and 16 months (95% CI 14.4–17.6) for the HGSOC group who underwent IDS, *p* = 0.897 and 0.874, respectively.

For the whole cohort of patients, superior survival rates were established in those patients in whom a complete surgical cytoreduction (CC 0–1) was achieved compared to in those with an incomplete cytoreduction (CC ≥ 2). The 5-year OS was 32.8% for the patients with a CC of 0–1versus 0% for the patients with a CC ≥ 2, *p* < 0.0001. The median OS for the LGSOC patients with a CC of 0–1was not reached: mean OS 64.8 months (95% CI 55.3.6–74.3) and median PFS 26 months (95% CI 8.7–43.3) versus HGSOC OS 42 months (95%CI 36.3–47.7) and PFS 18 months (95% CI 16–20) for those with a CC of 0–1, *p* = 0.012 and 0.016, respectively (Figure 4). The median OS and PFS for the LGSOC patients with a CC ≥ 2 were 30 months (95% CI 23–37) and 10 months (95 % CI 7.9–12.1) compared to 18 months (95%CI 9.2–26.8) and 11 months (95% CI 9.6–12.4) for the HGSOC group with a CC > 2, *p* = 0.746 and 0.329, respectively (Figure 4).

A superior OS was established in the entire cohort who received more complex surgery (SCS ≥ 4) when compared to those who had low surgical complexity (SCS 1–3). The 5-year OS was 44 % for patients with an SCS ≥ 4 versus 24 % for patients with an SCS 1–3, *p* < 0.002. The median OS for the LGSOC patients with an SCS ≥ 4 was not reached, with mean an OS of 66.5 (95% CI 55.3–77.7) and a median PFS of 36 (95% CI 25–47) months versus 50 (95%CI 37.7–62.3) and 21 (95%CI 15–26) months for those HGSOC patients with an SCS ≥ 4, *p* = 0.019 and 0.132, respectively (Figure 3). The median OS and PFS for the LGSOC patients with an SCS of 1–3 were 31 months (95% CI 14.5–47) and 14 months (95% CI 8.5–19.5) versus 35 months (95%CI 28.2–40.8) and 16 months (95%CI 14–18) for the HGSOC group with an SCS of 1–3, *p* = 0.743 and 0.823, respectively (Figure 5).

### 3.5. Multivariate Analysis of Surgical Characteristics

For the LGSOC patients, the favorable prognostic features for OS that were determined in the univariate analysis (*p* < 0.1) were PS, FIGO stage, PDS, CC 0–1, SCS ≥ 4, and CD < 3. The favorable prognostic features for the HGSOC patients were the same, with the exception of FIGO stage. In the multivariate analysis, the advantageous prognostic features for OS in the LGSOC patients were a CC of 0–1 (HR 62.4, 95% CI 6.8–567.9, *p* < 0.001) and an SCS ≥ 4 (HR 5.3, 95%CI 1.2–22.8, *p* = 0.024). For the HGSOC patients, the favorable prognostic features that were determined in the multivariate analysis were PDS (HR 1.8, 95% CI 1.1–2.9; *p* = 0.017) and a CC of 0–1 (HR 4.0, 95% CI 2.4–6.6, *p* = <0.001). Further details are displayed in Table 3.

## 4. Discussion

The present study demonstrates no clear survival advantage of chemotherapy in LGSOC patients. Despite the absence of adjuvant treatment in the majority of LGSOC patients, their 5-year OS and PFS were approximately twice that of the HGSOC patients. This analysis advances the evidence supporting a disparity in the clinical and surgical characteristics between LGSOC and HGSOC.

Reports of superior survival in LGSOC compared to in HGSOC have previously been confirmed [29]. Even with advanced stage disseminated disease, similar age, and performance status, the survival difference between LGSOC and HGSOC was substantial. Hence, we could not support that histology was less significant in the advanced stages of disease [30]. Despite the endorsement of adjuvant chemotherapy and/or endocrine therapy in advanced LGSOC by the National Comprehensive Cancer Network [31], there was no significant survival difference between our patients who received adjuvant chemotherapy and those who did not (median not reached, mean OS 61.5 months 95% CI 30.3–92.8 versus 65.3 95% CI 52.6–78.1 months, respectively). However, endocrine therapy and/or chemotherapy were routinely started in patients whose advanced LGSOC had progressed in our cohort of patients (82%). The limited response of LGSOC to systemic chemotherapy [8] is in support of our limited use of adjuvant chemotherapy. Our reported survival rates are in line with other series [10,32] although superior outcomes might be possible with alternative adjuvant treatments that were not explored in this context. A substantial number of LGSOC patients in our cohort received NACT, which may contradict the latter; however, only 38% (5/13) of those who received NACT had a partial radiological response, with two patients demonstrating a slight reduction in omental cake, and three showing a reduction in pleural effusion or ascites. In contrast, the other six patients had stable disease, and two women experienced radiological progression whilst on NACT. Although the use of NACT in LGSOC may be controversial, stable disease in 88% of the cases following NACT in LGSOC has been previously reported [9]. Therefore, in our cohort of patients, NACT was generally used to cover the time that was needed to optimize patients for an expected extended surgical cytoreductive procedure.

LGSOC remains a disease that primarily requires surgical treatment. A complete surgical cytoreduction should be the aim of surgeons who are operating on ovarian cancer in both the PDS as well as in the IDS settings. In this study, we used a residual disease classification (CC0/1/2/3) that is not commonly used to describe ovarian cancer debulking surgery but that is becoming standard practice in surgical oncology. It is classified by more discrete variables that aim to improve the precision of residual disease status reporting and has been used in other ovarian cancer survival analysis studies [33]. The satisfactory survival rates that were established in both the LGSOC and HGSOC patients who underwent a complete surgical cytoreduction (CC 0-1) in our study is in contrast with the poor prognosis of those patients who underwent an incomplete cytoreduction (CC ≥ 2). The multivariate analysis supported the notion that complete cytoreduction is an independent covariate for survival in LGSOC as well as in HGSOC. This agrees with previous studies on cytoreductive surgery in low-grade [34] and high-grade ovarian tumours [12]. Developing methods to predict surgical outcomes is important to identify who will benefit from maximal cytoreductive effort in the primary or interval surgical setting. Data mining technologies appear to be promising for non-invasive, clinically meaningful improvements in the prediction accuracy of pre-surgical patient selection [35]. Although this approach may not be yet validated in ovarian cancer, it has the potential to be more reflective than other intra-operative assessments such as the PCI index.

Patients with LGSOC had a lower pre-treatment CA125 level and more frequent calcified deposits on the CT scan when compared to the patients with HGSOC. This observation has been demonstrated in other studies [36,37]. Remarkably, in our population, the LGSOC and HGSOC patients were of a similar age at initial presentation. We could not confirm the reported younger age of the LGSOC patients compared to the HGSOC [10]. Variations in the population as well as in genetic predisposition may well explain this difference. LGSOC and HGSOC are separate entities of serous EOC and have different risk factors and distinctive clinical courses. They also have contrasting molecular characteristics: LGSOC is often characterized by gene mutations that are involved in the mitogen-activated protein kinase pathway, such as KRAS or BRAF mutations [38], in addition to PIK3CA driver mutations, which are linked to the AKT-mTOR pathway [39]. whilst HGSOC is associated with a greater frequency of P53 and BRCA1/2 mutations [40]. Furthermore, eostrogen and progesterone receptors are more commonly expressed in LGSOC than HGSOC, supporting the idea that endocrine therapy should maintained [41,42].

In addition to the aforementioned differences between LGSOC and HGSOC, there is also a clear disparity in surgical management. The finding that the LGSOC patients in our study required more complex surgical procedures (aiming at CC 0–1) when compared to those with HGSOC has, to our knowledge, not been reported before. The ramifications of more complex surgery in LGSOC were longer procedural time, more intra-operative blood loss, more HDU/ICU admissions, longer hospital stay, and consequentially, increased peri-/post-operative morbidity. This has been validated by a previous study that demonstrated a correlation between a higher PCI score and prolonged surgical time, increased post-op complications, and extended hospitalization [43]. Additionally, similar implications of more complex surgery that carries an increased risk of complications but that demonstrates a survival benefit by achieving complete surgical resection have been previously reported in unstratified advanced EOC [44]. Our study also demonstrated an independent association of more complex surgery with superior survival outcomes. This was confirmed by the multivariate analysis, which showed a survival benefit for more complex surgery in LGSOC in our study. However, we were unable to demonstrate this positive impact of more complex surgery on survival in our HGSOC patients, which was most likely due to the response of the relatively large number of patients to NACT in the HGSOC group. This is further supported by the higher PCI that was observed in the LGSOC patients compared to in the HGSOC patients. These observations indicate that we need to aim for a stratified, subtype-specific, approach for the treatment of advanced EOC, as suggested previously [45].

The findings of our study should not be overstated. Confounding variables are the relatively small number of patients with LGSOC who were recruited over a timeframe of 4 years and the non-randomized character of the study. Furthermore, two patients did demonstrate a radiological response to solid disease while on NACT, although this was restricted to a modest reduction in omental cake, thus supporting chemoresistance, and it does not confirm total futility in all patients. Equally, data on patients with LGSOC are not widely available, making randomization almost unachievable. We also acknowledge the fact that our 20% rate of primary cytoreductive surgery for advanced EOC patients falls short of our predetermined ambition.

## 5. Conclusions

Our study supports the consideration of omitting adjuvant systemic chemotherapy treatment for the management of advanced-stage LGSOC. The survival benefit of LGSOC compared to HGSOC is maintained despite the lack of chemotherapy, highlighting the significance of surgical treatment for this disease entity. Radical multi-visceral surgery that aims at complete surgical cytoreduction should be considered to be the primary treatment for LGSOC patients. A clear positive association between increased surgical complexity and superior survival outcomes in LGSOS was demonstrated. Nevertheless, increased operation time, longer hospital stays, increased HDU/ICU requirements, and increased post-operative morbidity resulting from this paradigm of surgical management should also be taken into consideration. LGSOC and HGSOC are different entities of serous EOC and consequently deserve different management. Therefore, we strongly recommend stratifying future studies in EOC to these separate subtypes. Prospective clinical trials should maintain an emphasis on systemic therapies that target specific LGSOC pathways such as MEK or PI3K inhibitors. Additional translational research is required to identify molecular predictors of response for LGSOC-targeted systemic treatment and endocrine therapy.

## Figures and Tables

**Figure 1 jcm-10-05927-f001:**
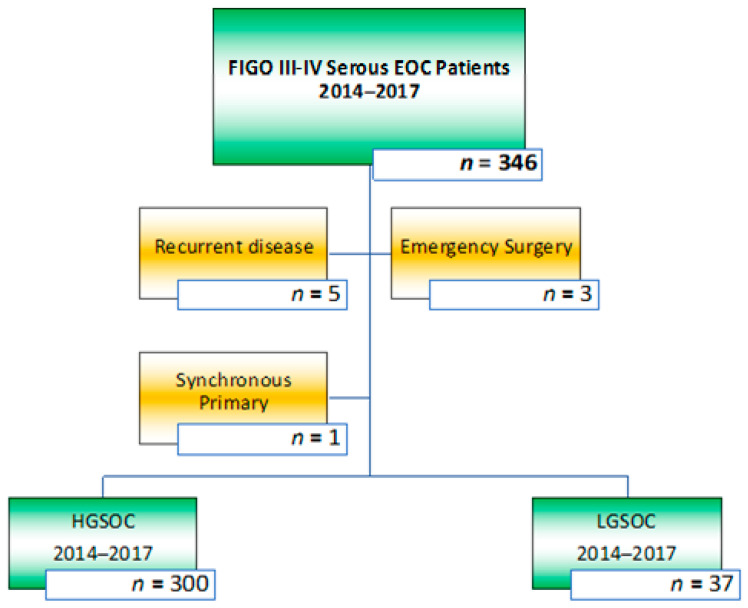
Inclusion and exclusion criteria for all patients with an advanced stage EOC who had cytoreductive surgery between January 2014 and December 2017. A total of 37 low-grade serous ovarian cancer (LGSOC) and 300 high-grade serous ovarian cancer (HGSOC) patients were included in the study.

**Figure 2 jcm-10-05927-f002:**
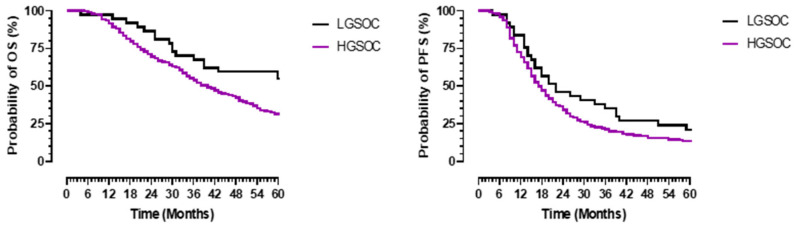
Overall (**left panel**) and progression-free (**right panel**) survival in patients with advanced-stage serous epithelial ovarian cancer according to grade. Survival is in months (X-axis), and probability of survival is in percentage (Y-axis). The black line represents patients with advanced low-grade serous cancer (LGSOC), and the purple line represents patients with high-grade serous ovarian cancer (HGSOC).

**Figure 3 jcm-10-05927-f003:**
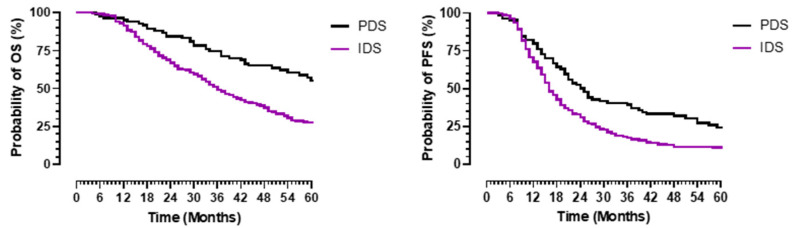
Overall (**left panel**) and progression-free (**right panel**) survival in patients with advanced-stage serous epithelial ovarian cancer according to surgical setting. Survival is in months (X-axis), and the probability of survival is in percentage (Y-axis). The black line represents advanced low- and high-grade serous cancer patients who underwent primary debulking surgery (PDS), and the purple line represents advanced low- and high-grade serous cancer patients who underwent interval debulking surgery following neoadjuvant chemotherapy.

**Figure 4 jcm-10-05927-f004:**
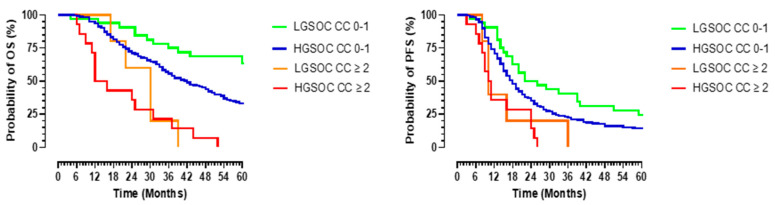
**The** overall (**left panel**) and progression-free (**right panel**) survival in patients with advanced-stage serous epithelial ovarian cancer according to completeness of cytoreduction. Survival is in months (X-axis), and the probability of survival is in percentage (Y-axis). Complete cytoreduction is represented by a CC of 0–1, and incomplete cytoreduction is represented by a CC ≥ 2. The green and orange lines represent patients with advanced low-grade serous cancer (LGSOC), and the blue and red lines represent patients with high-grade serous ovarian cancer (HGSOC).

**Figure 5 jcm-10-05927-f005:**
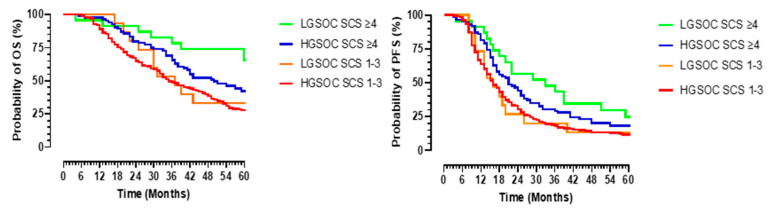
Overall (**left panel**) and progression-free (**right panel**) survival in patients with advanced stage serous epithelial ovarian cancer according to surgical complexity score (SCS). Survival is in months (X-axis), and probability of survival is in percentage (Y-axis). Low surgical complexity is represented by SCS 0–3, and medium to high surgical complexity is represented by SCS ≥ 4. The green and orange lines represent patients with advanced low-grade serous cancer (LGSOC), and the blue and red lines represent patients with high-grade serous ovarian cancer (HGSOC).

**Table 1 jcm-10-05927-t001:** Baseline characteristics of 337 advanced-stage (FIGO III-IV) epithelial serous ovarian cancer (EOC) patients who underwent cytoreductive surgery between the years 2014–2017. Numbers are shown as absolute numbers with percentage unless otherwise indicated.

	Low Grade Serous EOC 2014–2017	High Grade Serous EOC 2014–2017	*p*-Value
Patients	*n* = 37	*n* = 300	
Age (yrs) (Mean, SD)	61.3 ± 10.9	63.9 ± 10.2	0.164
Performance status (PS)			0.419
PS 0	20 (54.1%)	124 (41.3%)	
PS 1	12 (32.4%)	122 (40.7%)	
PS 2	3 (8.1%)	42 (14.0%)	
PS 3/4	2 (5.4%)	12 (4.0%)	
Pre-treatment CA125 (U/mL) (Median, Range)	122 (25–9657)	875 (13–28,600)	<0.0001
Pre-treatment Cytology/Histology			
Cytology	0 (0%)	2 (0.7%)	
Biopsy	37 (100%)	298 (99.3%)	
Pre-Treatment CT (Chest/Abdomen/Pelvis)			<0.0001
Calcified Deposits Present	29 (78.4%)	17 (5.7%)	
Absent Calcifications	8 (21.6%)	283 (94.3%)	
FIGO Stage			0.478
III A-B	7 (11.9%)	23 (13.9%)	
III C	24 (62.1%)	189 (58.8%)	
IV A-B	6 (26.0%)	88 (27.3%)	

FIGO: International Federation of Gynecology and Obstetrics.

**Table 2 jcm-10-05927-t002:** Surgical and treatment parameters of 337 advanced serous epithelial ovarian cancer (EOC) patients who underwent cytoreductive surgery. Numbers are absolute numbers with percentage unless otherwise indicated.

	Low-Grade Serous EOC 2014–2017	High-Grade Serous EOC 2014–2017	*p*-Value
**Patients**	*n* = 37	*n* = 300	
**Initial Treatment**			< 0.0001
Interval debulking surgery	13 (35.1%)	240 (80.0%)	
Primary debulking surgery	24 (64.9%)	60 (20.0%)	
**Peritoneal cancer index (PCI)** (Median, Range)	8 (2–21)	5 (1–24)	0.0216
Surgical Cytoreduction			0.8329
Complete (CC 0–1)	17 (73%)	264 (88.0%)	
Incomplete (CC ≥ 2)	10 (27%)	36 (12%)	
**Surgical Complexity Score (SCS)**			<0.0001
Low (1–3)	14 (37.8%)	213 (71.0%)	
Intermediate (4–7)	20 (54%)	74 (24.6%)	
High (8–18)	3 (8.1%)	13 (4.4%)	
**Operative time (minutes)** (Mean, SD)	207 ± 93	150 ± 63	<0.0001
**Intra-operative Blood Loss (cc)** (Mean, SD)	632 ± 329	478 ± 323	0.0019
**Post-operative Destination**			<0.0001
Regular Ward	23 (62.2%)	251 (83.7%)	0.0015
HDU/ICU	9 (4.9%)	72 (37.1%)	
**Length of Hospital Stay (days)** (Median, Range)	9 (4–30)	7 (3–68)	0.0002
**Peri-operative Morbidity (Clavien–Dindo)**			<0.0001
0–2	27 (73.0%)	280 (93.4%)	
3–4	10 (27.0%)	19 (6.3%)	
5	0 (0.0%)	1 (0.3%)	
**Adjuvant Treatment**			<0.0001
Platinum-based chemotherapy	10 (27 %)	297 (99%)	
Other (Chemo-)Therapy	0 (0.0%)	2 (0.7%)	
No Adjuvant Treatment	27 (73%)	1 (0.3%)	
**End of Treatment CA125 (U/mL)** (Median, Range)	17 (5–139)	13 (3–4019)	0.2152

**Table 3 jcm-10-05927-t003:** Multivariate analysis for improved overall survival in advanced low-grade serous ovarian cancer (LGSOC) and high-grade serous ovarian cancer (HGSOC). The covariates with *p* < 0.1 were analyzed by Cox’s proportional hazard regression controlled for performance status.

	Multivariate Analysis OS LGSOC			Multivariate Analysis OS HGSOC		
Covariates	HR	*p*	95% CI	HR	*p*	95% CI
**FIGO stage**						
III A-B	0.0001	0.9	0.0–1.1	0.9	0.83	0.38–2.2
III C	0.125	0.16	0.07–2.3	0.74	0.59	0.25–2.2
IV A	0.26	0.158	0.0–4.1	0.218	0.162	0.5–1.1
IV B	0.061	0.18	0.01–3.7	1.1	0.54	0.7–1.9
**Cytoreduction**						
Complete (CC 0–1)	62.3	<0.001	6.8–567.9	4.0	<0.001	2.4–6.6
**Surgical complexity score**						
>4	5.3	0.024	1.2–22.8	0.88	0.56	0.6–1.3
**Surgical setting**						
**Primary debulking**	3.2	0.16	0.6–16.6	1.8	0.017	1.1–2.9
**Clavien–dindo**						
0/1	0.0001	0.93	0–0.01–0.089	0.84	0.78	0.26–2.7
2	0.02	0.21	0.002–0.5	0.86	0.646	0.27–2.8
3	0.09	0.18	0.003–3	1.2	0.727	0.35–4.5
4	0.013	0.04	0.001–0.082	-	-	-

FIGO: International Federation of Gynecology and Obstetrics.

## Data Availability

Referenced in the text.

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
