# Peer review of "The Uncertain Benefit of Adjuvant Chemotherapy in Advanced Low-Grade Serous Ovarian Cancer and the Pivotal Role of Surgical Cytoreduction"

_jcm, 2021, doi:10.3390/jcm10245927_

Round 1
Reviewer 1 Report
Comprehensive genomic profiling is gradually becoming an inevitable part of the everyday oncology clinical practice, especially for rare tumors and tumors with low response rate to standard oncology therapy. The interpretation and optimal implementation of the results leads towards a more individualized and targeted approach to the patient.
LGSOC is generally an indolent, chemoresistant and possibly hormon-sensitive disease. LGSOC often harbors activating mutations of genes involved in the mitogen-activated protein kinase and PIK3Ca-AKT-mTOR pathways. Sometimes LGSOC has overexpression of insulin-like growth factor-1 and much lower frequency of p53 and BRCA gene mutations, compared with HGSOC.
Targeted therapies, primarily bevacizumab, MEK inhibitors and CDK inhibitors, alone or in combination with other antitumor therapy, are the focus of several ongoing and future clinical trials.
Author Response
Hello,
Many thanks for taking the time to review my paper and for your valued comments and suggestions.
You have made an excellent point. I have very much focussed on the surgical aspect of treatment without mentioning current research on proposed targeted systemic treatments. I have made amendments to reflect your comments in my discussion.
Reviewer 2 Report
Dear authors,
thank you for this interesting and well written manuscript, that will be of benefit for all that will deal with this tumor.
Please find some minor suggestions:
- line 45 please correct some typos (in in the neoadjuvant..)
In literature (10.1016/j.inat.2020.100668) is possibile to find report of metastases from ovarian cancer. Did you considered these patients? If yes, does these group of metastatic patients benefit from adjuvant chemotherapy? Please discuss.
Thank you.
Congratulations.
Author Response
Hello,
Many thanks for your taking the time to review my paper and for your valued comments and suggestions.
All of the patients included had stage III (local metastasis limited to the abdominal cavity) or IV ovarian cancer (distant metastasis). So we only included metastatic patients and LGSOC respond poorly to chemotherapy hence the focus on radical surgical resection. In the case report you have mentioned this lady also had primary radical brain surgery to remove her cancer rather than primary chemotherapy.
Regarding recurrent/relapsed ovarian cancer, LGSOC also respond poorly to chemotherapy (reference 9 in my manuscript; doi: 10.1016/j.ygyno.2009.03.001) so often if LGSOC recur they are reviewed to see if their recurrent disease is amenable to surgical resection first (as with the case report) and if not then our only option is chemotherapy (which can have limited response which is better than nothing) or endocrine therapy. One of the challenges is that there are no standard molecular/radiological marker for predicting which LGSOC patient will have a chemotherapy response, it is a case of trial and error.
I hope this answers your questions.
I have added '-' to line 45.
KR
Reviewer 3 Report
I am impressed of how much NACT you have, about 80% in high grade advanced ovarian cancer. What parameters do you use to decide to perform PDS or IDS?
It could be interesting to differentiate between the mean of PCI previous to NACT and preoperative in HGSOC, I assume that the data you add at the paper is the preoperative one.
According to that, your Low grade group has higher PCI than the high grade group, and because of that, the operation time is higher and you have more postoperative complications etc (as you comment in line 331 and after). I give you an article where it is demonstrated (A. Llueca, J. Escrig / EJSO 44 (2018) 163e169; doi: https://doi.org/10.1016/j.ejso.2017.11.003) if you want to use it.
Author Response
Hello,
Many thanks for taking the time to read and respond to my paper.
Yes, 80% NACT/IDS and 20% PDS, falls short of our units aspirations. Decision for IDS vs PDS is made at MDT (gynae-onc surgeons, medical oncologist, radiologist, pathologist and gynae-onc nurses) based on CT findings, physical examination, general health and performance status of the patient. We do not routinely perform laparoscopy to determine if resection is feasible.
Yes the PCI is calculated during the operation, it would be interesting to determine what difference NACT makes to PCI score.
Your article is interesting. It agrees that high PCI scores correlate to increased complications/hospital admission etc but that high PCI scores are associated with reduced survival outcomes, whereas higher PCI scores in LGSOC were associated with better survival outcomes. However this article does not identify which type of ovarian cancer, just 'advanced'. I wonder whether it is likely this article included mainly HGSOC which does not gain as much benefit from radical surgery/higher PCI scores than LGSOC does (albeit both benefit from residual tumour=0cm) and that our data included such a large proportion of HGSOC patients who had a positive pre-operative response from NACT. I have now included this as a reference.
Thanks again for your time. Have a lovely Christmas.